# Novel Predictive Strategy Using CA19-9 and Fecal Elastase Levels to Make Treatment Decisions for Resectable Pancreatic Cancer: A Retrospective Study

**DOI:** 10.3390/biomedicines13010062

**Published:** 2024-12-30

**Authors:** Hyung Sun Kim, Woojin Kim, Won-Gun Yun, Hye-Sol Jung, Youngmin Han, Mirang Lee, Wooil Kwon, Jin-Young Jang, Joon Seong Park

**Affiliations:** 1Pancreatobiliary Cancer Clinic, Department of Surgery, Gangnam Severance Hospital, Yonsei University College of Medicine, Seoul 03722, Republic of Korea; milk8508@yuhs.ac; 2Department of Preventive Medicine, Yonsei University College of Medicine, Seoul 03722, Republic of Korea; laraed04@gmail.com; 3Korea Medical Institute, Seoul 04522, Republic of Korea; 4Department of Surgery and Cancer Research Institute, Seoul National University Hospital, Seoul National University College of Medicine, Seoul 03080, Republic of Korea; 5Division of Hepatobiliary and Pancreatic Surgery, Department of Surgery, Asan Medical Center, Seoul 05505, Republic of Korea

**Keywords:** resectable pancreatic cancer, CA19-9, fecal elastase-1, preoperative tumor marker, cutoff value

## Abstract

**Background**: Carbohydrate antigen 19-9 (CA19-9) is used as a marker to predict recurrence and survival of patients with pancreatic ductal adenocarcinoma (PDAC). Recently, fecal elastase-1 (FE-1) has been shown to correlate with prognosis in patients with PDAC. **Method**: A total of 536 patients who underwent curative intent surgery between 2010 and 2019 were included in the study. The cutoff points of preoperative CA19-9 and FE-1 levels were extracted from the Youden index and previous studies. Cox proportional hazard models were used to investigate the association between preoperative tumor marker levels and survival after surgery. **Results**: Patients with CA19-9 ≥ 385 had more advanced T-/N-stages and lower survival rates compared to those with CA19-9 < 385. Multivariate Cox analyses demonstrated that combining preoperative tumor markers was associated with worse 3-year overall survival (both CA19-9 and FE-1 low, HR = 1.41, *p* = 0.044; both high, HR = 1.44, *p* = 0.047; CA19-9 high and FE-1 low, HR = 2.00, *p* < 0.001; and *p* for trend < 0.001). The same trend was confirmed in the analysis with recurrence-free survival. **Conclusions:** This study presents a new predictive strategy using combined CA19-9 and FE-1 levels to determine the treatment for resectable pancreatic cancer.

## 1. Introduction

Pancreatic ductal adenocarcinoma (PDAC) is fatal. Recent advances have facilitated early screening and diagnosis of PDAC; however, the overall 5-year survival rate of patients with pancreatic cancer has not increased [1,2]. Even after curative surgery, the early recurrence rate is high and it is difficult to ensure an R0 resection rate of 100% [3,4,5]. Early diagnosis using effective methods is crucial to reduce mortality and improve survival rates [6]. In borderline resectable pancreatic cancer, the National Comprehensive Cancer Network (NCCN) guidelines suggest that neoadjuvant treatment is beneficial [7]. However, it is not certain whether neoadjuvant treatment or upfront surgery is effective for resectable pancreatic cancers. Some institutions have started including neoadjuvant treatment for resectable pancreatic cancer based on the previously demonstrated effects of neoadjuvant treatment in patients with borderline resectable pancreatic cancer [8]. However, a predictive marker to determine whether a treatment is more useful for resectable pancreatic cancer is not available.

To date, the most useful prognostic marker in patients with pancreatic cancer is CA19-9, and there is no consensus on the cutoff value of CA19-9 for deciding treatment. Various cutoff values have been suggested for CA19-9 levels in other studies. According to the standards presented in the international guidelines for borderline resectable pancreatic cancer, the biological definition of BR-PDAC was determined to be preoperative CA19-9 levels of 500 IU/mL using resection rate and survival time [9].

There are many disadvantages of using a single marker. CA19-9 can be difficult to measure due to its dependence on Lewis antigens and is also influenced by inflammation at the time of measurement [10]. It has the disadvantage of requiring invasive procedures, making frequent measurements challenging. Given these limitations as a single marker, we believe that combining it with other risk factors to create a new marker holds significant potential.

Accordingly, we suggest another marker, fecal elastase (FE-1), to develop a new predictive strategy for PDAC. Fecal elastase (FE-1) is a well-known indicator of severe pancreatitis [11,12], and FE-1 reflects the degree of pancreatic fibrosis. Pancreatic fibrosis has been thought to play an important role in carcinogenesis [13,14,15]. Moreover, in our previous study, we reported that reduced FE-1 levels were a significantly unfavorable independent prognostic factor of disease-free survival for patients with PDAC after curative resection [16]. Fibrosis of the pancreatic tissue results in the loss of pancreatic functional tissue and inflammation [16]. An association between pancreatic cancer and inflammation has been reported [17,18,19,20,21,22,23]. FE-1 levels can be easily measured in patients. Therefore, FE-1 is a useful predictive marker [24]. We examined the data for two markers, viz. CA19-9 and FE-1, in this study to develop a new predictive strategy and to determine the direction of treatment for resectable pancreatic cancer.

## 2. Materials and Methods

### 2.1. Study Participants and Design

We conducted a retrospective study using data from 857 patients with pancreatic cancer enrolled in two hospitals, Gangnam Severance Hospital and Seoul National University Hospital, in the Republic of Korea from 5 January 2010 to 31 December 2019.

Among them, we excluded patients without information about preoperative tumor marker levels (n = 45), those who received neoadjuvant (n = 169), those with preoperative CA19-9 levels < 9.0 (Lewis antibody-negative patients) (n = 98), those with missing covariates (n = 3), and those with R2 resection (n = 6).

Finally, we considered the data of 536 patients (Figure A1). The endpoints of this study were 3-year overall survival (OS) and 1-year recurrence-free survival (RFS). Study enrollment date was defined as the day of surgery. All patients who agreed to participate in the study signed an informed consent form before enrollment. This study was registered with ClinicalTrials.gov (NCT05923567) and approved by the Institutional Review Board (IRB) of Gangnam Severance Hospital (No. 3-2020-0522; approval date: 24 February 2022).

### 2.2. Measurement for Preoperative FE-1

For FE-1 measurement, stool specimens were obtained 4 days preoperatively. Liquid stools were excluded from the examination due to falsely low FE-1 levels resulting from dilution. A commercially available enzyme-linked immunosorbent assay (ELISA) kit (Schebo Biotech AG, Giessen, Germany) was used to measure FE-1 concentration. The assays also shared independently established but equivalent reference intervals of <100, 100–200, and >200 µg FE-1/g stool for severe pancreatic insufficiency, moderate insufficiency, and normal pancreatic function, respectively.

### 2.3. Statistical Analyses

#### 2.3.1. Defining an Optimal Cutoff Value for Preoperative Tumor Markers

Cutoff values for each of the two preoperative tumor markers were determined and studied simultaneously to classify the subjects into four groups. We used the preoperative CA19-9 value with normalized serum bilirubin level. First, the cutoff value of preoperative CA19-9 was defined by the Youden index (Youden’s J statistic), which is defined as [sensitivity  +  specificity − 1]. Receiver operating characteristic (ROC) analysis was performed, and the area under the curve (AUC) was calculated to identify the optimal cutoff for discriminating death within 3 years of surgical therapy for resectable pancreatic cancer. The cutoff point was the threshold that maximized the J statistic, which was the highest combination of sensitivity and specificity. Second, the FE-1 cutoff value in this study was set to 100 µg FE-1/g stool (<100 and ≥100 µg FE-1/g), with reference to previous studies [11,12].

#### 2.3.2. General Characteristics of Patient Data Groups

The baseline characteristics of the four groups of patient data were defined based on the cutoff values of two preoperative tumor markers, CA19-9 and FE-1. Age was divided into four groups by quartiles. Data were presented as frequencies with percentages for categorical data and averages with standard deviation (SD) (median with interquartile range for non-normally distribution) for continuous data. Differences between groups were compared using the chi-square or Fisher’s exact test for categorical variables and *t*-test or ANOVA (Wilcoxon rank-sum test or Kruskal–Wallis test for non-normally distribution) for continuous variables.

#### 2.3.3. Kaplan–Meier and Cox Regression Analysis

Survival curves were obtained using the Kaplan–Meier method (log-rank test). First, we compared the survival of the following four groups: CA19-9 below the cutoff and FE-1 above the cutoff value; both FE-1 and CA19-9 below the cutoff values; both FE-1 and CA19-9 above the cutoff values; and CA19-9 above the cutoff and FE-1 below the cutoff value. Additionally, to confirm the difference in survival rate according to preoperative cancer marker levels in early stage, the same analysis was performed in subjects whose preoperative cancer size was 2 cm or less. Next, the survival of the two groups, defined as preoperative FE-1 levels in the low CA19-9 (below the cutoff values) group and the high CA19-9 (above the cutoff values) group, respectively, was also compared.

Univariate and multivariate analyses were performed using the Cox proportional hazard regression model to determine the effects of different variables on survival. The multivariate model was adjusted for age (<58, ≥58 and <66, ≥66 and <73, or ≥73 years), sex (male or female), T classification (T1, T2, or T3/T4), N classification (N0, N1, or N2), perineural invasion (negative or positive), lymphovascular invasion (negative or positive), histologic differentiation (well, moderate, or poorly/undifferentiated), resection margin status (R0 or R1), and combined CA19-9 and FE-1 levels as covariates.

### 2.4. Additional Analysis

We also performed supplemental analyses to examine the robustness of the previous findings. Additional analyses investigated the combination effects of two preoperative tumor maker levels on survival for longer times: 5-year OS and 3-year RFS.

All statistical analyses were performed using the SAS software (version 9.4; SAS Institute Inc., Cary, NC, USA) and R version 4.2.1 (R Project for Statistical Computing). *p*-values <0.05 (two-tailed) were considered statistically significant.

## 3. Results

### 3.1. Clinical Characteristics of Patients with Pancreatic Cancer

Table 1 shows the characteristics of patients enrolled in this study. Among the enrolled patients, the mean age was 65.4 years; the study population included 318 men (59.3%) and 218 women (40.7%). Taking T staging into consideration, 67 (12.5%) patients had T1 stage disease, 346 (64.6%) had T2 stage disease, 113 (21.1%) had T3 stage disease, and 10 (1.8%) had T4 stage disease. With respect to N staging, 200 (37.3%) patients had N0 stage disease, 239 (44.6%) had N1 stage disease, and 97 (18.1%) had N2 stage disease. Perineural invasion was observed in 67 (12.5%) patients. Lymphovascular invasion was observed in 282 (52.6%) patients. Most of the patients showed moderate differentiation (n = 402, 75%) and R0 resection (n = 462, 86.2%).

### 3.2. Defining the Cutoff Value of CA19-9

The calculation for CA19-9 cutoff was based on data from a total of 536 patients. ROC curve analysis indicated a critical value of 385.11 to discriminate 3-year OS with the highest Youden index (sensitivity = 0.42, specificity = 0.75). The area under the ROC curve (AUC) was 0.61 (95% confidence interval [CI] 0.56–0.66) (Figure A2). In this study, the cutoff value was set at 385.0 for the convenience of interpreting the results.

### 3.3. Clinical Characteristics of Four Groups of Patients Using CA19-9 Cutoff and FE-1 Values

The baseline characteristics of groups divided by preoperative tumor marker levels are presented in Table 2 and Table A1. Table 2 shows the clinical characteristics of the four groups—CA19-9 below and FE-1 above the cutoff values (Group 1, reference group), both below (Group 2), both above (Group 3), and CA19-9 above and FE-1 below (Group 4) the cutoff values. The results showed significant differences in the T stage, N stage, and lymphovascular invasion. Advanced T and N stages were seen predominantly in Group 4. There were significant differences between groups in T and N stages, lymphovascular invasion, and resection margins. Four groups of patients showed significant differences in OS and RFS.

### 3.4. Kaplan–Meier Estimates of Overall Survival and Recurrence-Free Survival

The median OS of patients in Groups 1 to 4 was 25.7, 21.8, 18.3, and 17.1 months, respectively (*p* = 0.004); the OS rate at 3 years was 63.1% and the RFS rate at 1 year was 64.9%. The median RFS of patients in Groups 1 to 4 was 14.6, 14.1, 9.0, and 7.7 months, respectively (*p* < 0.001); the OS rate at 3 years was 31.2% and the RFS rate at 1 year was 37.5% (Table A2 and Table 2).

Kaplan–Meier survival analysis indicated that patients in Group 4 had the highest death/or recurrence rates while patients in Group 1 had the lowest rates. There were statistically significant differences in 3-year OS (*p* < 0.001), 5-year OS (*p* < 0.001), 1-year RFS (*p* < 0.001), and 3-year RFS (*p* < 0.001) rates between four groups divided by combining CA19-9 and FE-1 (Figure 1, Figure A3, and Figure A4). Among patients with CA19-9 < 385, the 3-year OS was shorter in patients with FE-1 < 100 (*p* = 0.056) (Figure A5 (left)). Among patients with CA19-9 ≥ 385, the 3-year (*p* = 0.053) and 5-year OS (*p* = 0.033) rates were shorter in the group with FE-1 < 100 (Figure A5 (right)) and Figure A6 (right)). There was no significant difference in RFS by FE-1 for both groups with preoperative CA19-9 less than or greater than 385 (Figure A7 and Figure A8).

In patients with small-sized tumors (≤2 cm), the OS and RFS rates in the four groups were significantly different (*p* < 0.001) (Figure 2). The patients included in Group 4 especially had the worst prognosis in terms of recurrence-free survival and overall survival. All patients (five in total) in this group had a recurrence of pancreatic cancer before 24 months and died before 33 months (Figure A9 and Figure A10).

### 3.5. Prognostic Impact of Clinicopathologic Features in Pancreatic Cancer

Upon univariate analysis, advanced T stage and N stage, perineural invasion, moderate to poor differentiation, high CA19-9 levels, and low FE-1 levels were identified as independent factors for poor recurrence-free survival. Old age, advanced T stage and N stage, perineural invasion, moderate to poor cellular differentiation, high CA19-9, and low FE-1 levels were identified as factors for poor overall survival (Table 3).

In multivariate analysis, advanced T stage and N stage, moderate to poor differentiation, high CA19-9 levels, and low FE-1 levels were identified as independent factors for poor recurrence-free survival. Old age, advanced N stage, moderate to poor cellular differentiation, high CA19-9, and low FE-1 levels were identified as factors for poor overall survival (Table 3).

The hazard ratio (HR) for OS and RFS was significantly associated with combined preoperative CA19-9 and FE-1 levels after data were controlled for age, sex, T stage, N stage, perineural invasion, lymphovascular invasion, and resection margin (Table 3). Compared to the group with CA19-9 below and FE-1 above the cutoff values, it was observed that the HR for 3-year OS gradually increased in Group 2 (HR = 1.41; 95% confidence interval (CI): 1.01, 1.97; *p* = 0.044), Group 3 (HR = 1.44; 95% CI: 1.01, 2.06; *p* = 0.047), and Group 4 (HR = 2.00; 95% CI: 1.40, 2.85; *p* < 0.001; *p* for trend < 0.001). HR for 5-year OS also showed a similar tendency, but no statistical significance was observed in Group 2 (Table A3). Compared to Group 1, the HR for 1-year RFS was higher in Group 3 (HR = 1.76; 95% CI: 1.24, 2.51; *p* = 0.002) and Group 4 (HR = 1.88; 95% CI: 1.30, 2.51; *p* = 0.002). Although the HR of Group 2 (HR = 0.89; 95% CI: 0.61, 1.29; *p* = 0.531) was not significant, it was confirmed that it gradually increased from Group 2 to Group 4 (*p* for trend < 0.001). A similar trend was observed for HR in the 3-year RFS (Table A4).

## 4. Discussion

This study presents a new predictive strategy using combined CA19-9 and FE-1 levels to determine the treatment for resectable pancreatic cancer. Our study showed that preoperative CA19-9 and FE-1 levels can be utilized as convenient prognostic indicators for survival outcomes in PDAC patients.

Patients with pancreatic cancer have high rates of recurrence and metastasis. CA19-9 is an effective prognostic marker for pancreatic cancer. In addition to CA19-9, there are various other prediction tools, such as PET-CT scans [25]. Despite this, CA19-9 levels are the most commonly used prognostic markers for pancreatic cancer as they can be easily determined and used for immediate prediction of the disease status [26,27,28,29,30,31]; however, they are affected by the systemic condition or inflammatory status of the patients [32]. In our study, patients with CA19-9 levels <9 U/mL were excluded since they were more likely to be negative for the Lewis antigen [33,34]. This may have influenced the results. Moreover, we used FE-1, which has been described in a previous study as a significant predictive marker [16].

Another study reported that the combined use of CA19-9 and C-reactive protein (CRP) levels is a novel prognostic strategy for pancreatic ductal adenocarcinoma [35]. This study suggested that CRP levels reflect the patient’s systemic reaction to the tumor. However, CRP showed all other systemic inflammatory responses, and it was difficult to consider it as a specific marker for pancreatic cancer. By using these two simple markers, CA19-9 and FE-1, we developed a novel predictive strategy in patients with PDAC who had undergone curative resection. The criteria for neoadjuvant treatment or upfront surgery for resectable PDAC have not yet been clearly determined. Our novel strategy will be helpful in determining personalized treatment considering the characteristics of each patient.

The biological definition of BR-PDAC was determined to be preoperative CA19-9 levels of 500 IU/mL. However, the cutoff value (CA19-9 = 385), based on the patient’s overall survival, was also established in our study. Similarly, the CA19-9 cutoff value of 385 IU/mL observed in our study showed a difference in survival at a high value. The CA19-9 values of patients with normalized bilirubin (total bilirubin < 3 mg/dL) were used in the present study. This is the reason why the cutoff value in our study was slightly different from the international guideline. Additionally, small-sized tumors (tumor size < 2 cm) were also analyzed to confirm the results for patients with truly resectable tumors. In general, small-sized tumors (T1 stage) are classified as early-stage tumors, and most small-sized tumors are resectable. We also confirmed the difference in survival curves for patients with these tumors, which means that the tumor size can be used as a reliable prognostic factor in cases with resectable tumors.

In our study, if FE-1 was used, even when the CA19-9 levels were low (CA19-9 <385), the prognosis could be predicted by dividing the group in detail. This may be more useful in determining treatment regimes, such as neoadjuvant treatment.

We evaluated a relatively large number of patients who underwent surgery for pancreatic cancer. To collect a sufficient amount of data, researchers spent 10 years building registries at two of the country’s largest medical institutions. Next, we used two markers that can be easily collected from patients. This makes it easier to utilize research results or conduct further research. Nevertheless, this study had some limitations. First, this was a retrospective multicenter study. Therefore, there were several differences in the treatment of resectable pancreatic cancer over the course of time and the treatment policies of each institution. Second, we analyzed cases of resectable pancreatic cancer; however, it was unclear whether all of these tumors were resectable, based on preoperative images. Third, the number of patients with pancreatic tumors smaller than 2 cm is generally limited. However, as the study is ongoing, further research may be possible as the patient population increases. Moreover, it is difficult to generalize the results of this study to larger populations, or to populations with different characteristics. We divided the patients into subgroups with CA19-9/FE-1 with our data, and it is necessary to validate this with a different data set. Further research is required for external validation in the future.

## 5. Conclusions

Our results showed that the prognosis was poor in patients with high CA19-9. Using FE-1 in patients with low CA19-9 and small tumors (T1), we predicted the precise prognosis and performed subgrouping. Our analyses of the correlation between survival outcomes and CA19-9 and FE-1 levels in patients with PDAC revealed that these values were accurate and convenient prognostic indicators of PDAC. In addition, this study provides a basis for the treatment of resectable pancreatic cancer using CA19-9 and FE-1 as predictive markers.

## Figures and Tables

**Figure 1 biomedicines-13-00062-f001:**
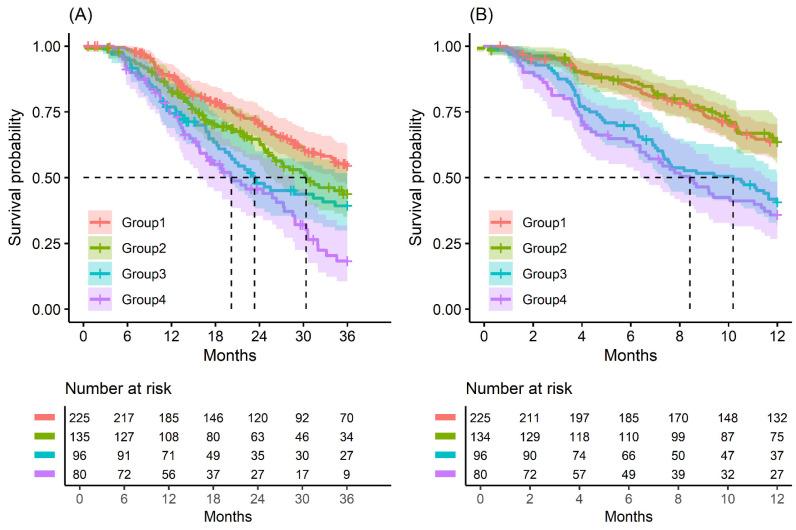
Kaplan–Meier survival curve by groups on based preoperative CA19-9 and FE-1 level. (**A**) 3-year overall survival (*p* < 0.001) and (**B**) 1-year recurrence free survival (*p* < 0.001). Group 1: CA19-9 below and FE-1 above the cutoff values; Group 2: both CA19-9 and FE-1 below the cutoff values; Group 3: both CA19-9 and FE-1 above the cutoff values; and Group 4: CA19-9 above and FE-1 below the cutoff values. The median survival time for each group, the time at which survival probability S(t) is 0.5, is indicated by dashed lines.

**Figure 2 biomedicines-13-00062-f002:**
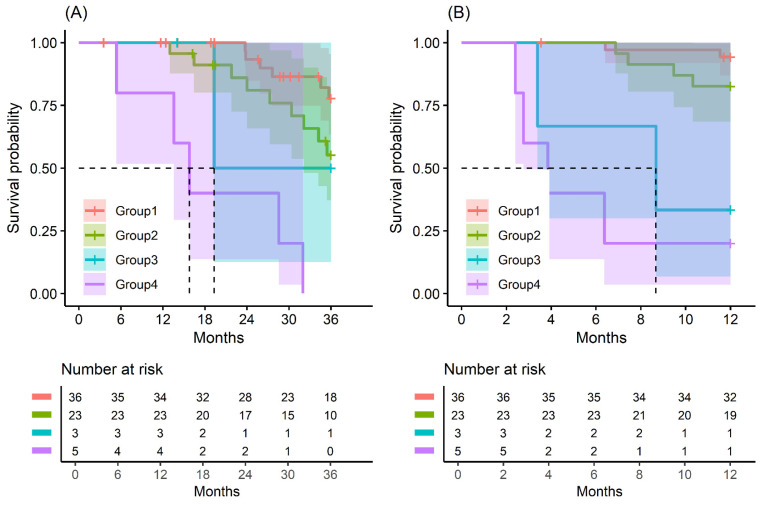
Kaplan–Meier survival curve by groups on based preoperative CA19-9 and FE-1 level on patients with tumor size ≤ 2 cm. (**A**) 3-year overall survival (*p* < 0.001) and (**B**) 1-year recurrence free survival (*p* < 0.001). Group 1: CA19-9 below and FE-1 above the cutoff values; Group 2: both CA19-9 and FE-1 below the cutoff values; Group 3: both CA19-9 and FE-1 above the cutoff values; and Group 4: CA19-9 above and FE-1 below the cutoff values. The median survival time for each group, the time at which survival probability S(t) is 0.5, is indicated by dashed lines.

**Table 1 biomedicines-13-00062-t001:** Clinical characteristics of the patients’ groups with pancreatic cancer (n = 536).

Variables	Mean/n (SD/%)
Age, mean (SD)	65.4 (9.6)
Age, n (%)	
<58	115 (21.5)
≥58 and <66	135 (25.2)
≥66 and <73	144 (26.9)
≥73	142 (26.5)
Sex, n (%)	
Male	318 (59.3)
Female	218 (40.7)
T classification, n (%)	
T1	67 (12.5)
T2	346 (64.6)
T3	113 (21.1)
T4	10 (1.8)
N classification, n (%)	
N0	200 (37.3)
N1	239 (44.6)
N2	97 (18.1)
Perineural invasion, n (%)	
Negative	469 (87.5)
Positive	67 (12.5)
Lymphovascular invasion, n (%)	
Negative	254 (47.4)
Positive	282 (52.6)
Histologic differentiate, n (%)	
Well	48 (9.0)
Moderate	402 (75.0)
Poorly or undifferentiated	79 (14.7)
Unknown	7 (1.3)
Resection margine, n (%)	
R0	462 (86.2)
R1	74 (13.8)

Abbreviations: SD, standard deviation.

**Table 2 biomedicines-13-00062-t002:** Clinical characteristics of four groups of the patients using CA19-9 cutoff and FE-1 values.

Variables	CA19-9 < 385 and FE-1 ≥ 100 (n = 225)	CA19-9 < 385 and FE-1 < 100 (n = 135)	CA19-9 ≥ 385 and FE-1 ≥ 100 (n = 96)	CA19-9 ≥ 385 and FE-1 < 100 (n = 80)	*p* Value
Age, mean (SD)	65.4 (9.9)	64.9 (10.4)	66.4 (9.0)	64.8 (8.3)	0.642
Age, n (%)					0.505
<58	47 (20.9)	33 (24.4)	16 (16.7)	19 (23.8)	
≥58 and <66	58 (25.8)	31 (23.0)	27 (28.1)	19 (23.8)	
≥66 and <73	59 (26.2)	34 (25.2)	23 (24.0)	28 (35.0)	
≥73	61 (27.1)	37 (27.4)	30 (31.2)	14 (17.5)	
Sex, n (%)					0.126
Male	130 (57.8)	89 (65.9)	49 (51.0)	50 (62.5)	
Female	95 (42.2)	46 (34.1)	47 (49.0)	30 (37.5)	
T classification, n (%)					0.002
T1	36 (16.0)	23 (17.0)	3 (3.1)	5 (6.3)	
T2	152 (67.6)	83 (61.5)	61 (63.5)	50 (62.5)	
T3	36 (16.0)	25 (18.5)	30 (31.3)	22 (27.5)	
T4	1 (0.4)	4 (3.0)	2 (2.1)	3 (3.8)	
N classification, n (%)					0.001
N0	97 (43.1)	60 (44.4)	27 (28.1)	16 (20.0)	
N1	97 (43.1)	51 (37.8)	48 (50)	43 (53.8)	
N2	31 (13.8)	24 (17.8)	21 (21.9)	21 (26.3)	
Perineural invasion, n (%)					0.086
Negative	193 (85.8)	113 (83.7)	89 (92.7)	74 (92.5)	
Positive	32 (14.2)	22 (16.3)	7 (7.3)	6 (7.5)	
Lymphovascular invasion, n (%)					0.001
Negative	94 (41.8)	58 (43.0)	49 (51.0)	53 (66.3)	
Positive	131 (58.2)	77 (57.0)	47 (49.0)	27 (33.8)	
Histologic differentiate, n (%)					0.346
Well	23 (10.2)	15 (11.1)	8 (8.3)	2 (2.5)	
Moderate	171 (76.0)	94 (69.6)	74 (77.1)	63 (78.8)	
Poorly or undifferentiated	29 (12.9)	22 (16.3)	14 (14.6)	14 (17.5)	
Unknown	2 (0.9)	4 (3.0)	0 (0.0)	1 (1.2)	
Resection margine, n (%)					0.017
R0	202 (89.8)	110 (81.5)	87 (90.6)	63 (78.8)	
R1	23 (10.2)	25 (18.5)	9 (9.4)	17 (21.2)	
Event, n (%)					
Death within 3-year	83 (36.9)	63 (46.7)	51 (53.1)	55 (68.8)	<0.001
Recurrence within 1-year	79 (35.1.2)	46 (34.1)	56 (58.4)	50 (65.2)	<0.001

Abbreviations: CA19-9, carbohydrate antigen 19-9; FE-1, fecal elastase-1; SD, standard deviation.

**Table 3 biomedicines-13-00062-t003:** Prognostic impact of clinicopathologic features in pancreatic cancer.

Variables	Univariate Cox Proportional Model						Multivariate Cox Proportional Model					
	Event = 3-Year OS			Event = 1-Year RFS			Event = 3-Year OS			Event = 1-Year RFS		
	HR (95% CI)	*p* Value	*p* Trend	HR (95% CI)	*p* Value	*p* Trend	HR (95% CI)	*p* Value	*p* Trend	HR (95% CI)	*p* Value	*p* Trend
Age												
<58	1	ref.	0.002	1	ref.	0.557	1	ref.	0.001	1	ref.	0.797
≥58 and <66	0.80 (0.55, 1.18)	0.267		0.88 (0.60, 1.28)	0.499		0.79 (0.53, 1.17)	0.234		0.82 (0.55, 1.2)	0.304	
≥66 and <73	1.01 (0.70, 1.46)	0.959		0.93 (0.64, 1.36)	0.716		1.01 (0.70, 1.46)	0.963		0.85 (0.58, 1.24)	0.391	
≥73	1.61 (1.14, 2.29)	0.008		1.06 (0.73, 1.54)	0.750		1.68 (1.18, 2.41)	0.004		1.03 (0.70, 1.50)	0.893	
Sex												
Male	1	ref.		1	ref.		1	ref.		1	ref.	
Female	1.01 (0.78, 1.30)	0.968		0.99 (0.76, 1.29)	0.945		0.98 (0.76, 1.27)	0.868		1.04 (0.79, 1.37)	0.783	
T classification												
T1	1	ref.	0.002	1	ref.	<0.001	1	ref.	0.439	1	ref.	0.085
T2	2.22 (1.39, 3.53)	0.001		2.94 (1.63, 5.29)	<0.001		1.50 (0.92, 2.44)	0.109		2.02 (1.09, 3.74)	0.025	
T3/4	2.46 (1.48, 4.07)	0.001		3.87 (2.09, 7.18)	<0.001		1.42 (0.83, 2.44)	0.201		2.13 (1.11, 4.09)	0.024	
N classification												
N0	1	ref.	<0.001	1	ref.	<0.001	1	ref.	<0.001	1	ref.	0.007
N1	1.66 (1.23, 2.24)	0.001		1.74 (1.28, 2.37)	<0.001		1.39 (1.01, 1.92)	0.046		1.40 (1.00, 1.96)	0.049	
N2	3.14 (2.24, 4.41)	<0.001		2.25 (1.55, 3.25)	<0.001		2.33 (1.60, 3.39)	<0.001		1.75 (1.16, 2.64)	0.007	
Perineural invasion												
Negative	1	ref.		1	ref.		1	ref.		1	ref.	
Positive	0.42 (0.26, 0.67)	<0.001		0.61 (0.39, 0.97)	0.035		0.67 (0.41, 1.10)	0.113		1.11 (0.68, 1.83)	0.671	
Lymphovascular invasion												
Negative	1	ref.		1	ref.		1	ref.		1	ref.	
Positive	0.60 (0.47, 0.77)	<0.001		0.61 (0.47, 0.80)	<0.001		0.84 (0.64, 1.10)	0.198		0.84 (0.63, 1.12)	0.237	
Histologic differentiate												
Well	1	ref.	0.001	1	ref.	<0.001	1	ref.	0.001	1	ref.	<0.001
Moderate	2.36 (1.35, 4.15)	0.003		3.61 (1.69, 7.68)	0.001		1.97 (1.11, 3.50)	0.021		3.03 (1.41, 6.53)	0.005	
Poorly or undifferentiated	3.02 (1.62, 5.63)	0.001		6.75 (3.05, 14.92)	<0.001		2.85 (1.51, 5.36)	0.001		6.23 (2.79, 13.92)	<0.001	
Resection Margine, N (%)												
R0	1	ref.		1	ref.		1	ref.		1	ref.	
R1	1.25 (0.88, 1.78)	0.220		1.28 (0.89, 1.85)	0.182		0.97 (0.67, 1.41)	0.876		1.20 (0.81, 1.76)	0.367	
Preoperative CA19-9 and FE-1												
CA19-9 < 385 and FE-1 ≥ 100	1	ref.	<0.001	1	ref.	<0.001	1	ref.	<0.001	1	ref.	<0.001
CA19-9 < 385 and FE-1 < 100	1.40 (1.01, 1.95)	0.046		1.00 (0.69, 1.45)	0.989		1.41 (1.01, 1.97)	0.045		0.88 (0.61, 1.29)	0.522	
CA19-9 ≥ 385 and FE-1 ≥ 100	1.72 (1.21, 2.44)	0.002		2.08 (1.47, 2.93)	<0.001		1.45 (1.01, 2.07)	0.045		1.75 (1.23, 2.50)	0.002	
CA19-9 ≥ 385 and FE-1 < 100	2.48 (1.75, 3.49)	<0.001		2.43 (1.70, 3.48)	<0.001		2.00 (1.40, 2.86)	<0.001		1.88 (1.29, 2.72)	0.001	

Abbreviations: OS, overall survival; RFS, recurrence-free survival; HR, hazard ratio; CI, confidence interval; CA19-9, carbohydrate antigen 19-9; FE-1, fecal elastase-1.

## Data Availability

The original contributions presented in this study are included in the article. Further inquiries can be directed to the corresponding author.

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
