# Peer review of "Novel Predictive Strategy Using CA19-9 and Fecal Elastase Levels to Make Treatment Decisions for Resectable Pancreatic Cancer: A Retrospective Study"

_biomedicines, 2024, doi:10.3390/biomedicines13010062_

Round 1

Reviewer 1 Report

Comments and Suggestions for Authors

This study addresses an important clinical question regarding prognostic markers in pancreatic cancer, combining CA19-9 and fecal elastase (FE-1) for treatment decisions. In conclusion, this study presents a new predictive strategy using combined CA19-9 and FE-1 levels to determine the treatment for resectable pancreatic cancer. I have following comments:

1) The cutoff values for CA19-9 (385 IU/mL) and FE-1 (100 µg/g) are pivotal to the analysis but appear derived primarily from internal data or prior literature. Clarify how confounders were adjusted in the multivariate analysis. Did the team perform sensitivity analyses to ensure the robustness of cutoff values (e.g., CA19-9 at 385)?

2) The manuscript does not mention whether the proportional hazards (PH) assumption was tested for the Cox models.

3) Multivariate analysis combines CA19-9, FE-1, and other clinicopathological variables, but multicollinearity between these predictors is not addressed.

4) Differences in baseline characteristics between the four groups based on CA19-9 and FE-1 cutoffs are mentioned (e.g., age, tumor stage). However, it is unclear if these differences were adjusted in survival analyses.

5) Subgroup analyses, particularly for small-sized tumors (≤2 cm), may suffer from limited sample size.

6) The area under the curve (AUC) for CA19-9 is reported as 0.61, which is modest and suggests limited discriminatory ability.

7) Numerous p-values are presented without adjustment for multiple comparisons.

8) Hazard ratios (HRs) are provided, but confidence intervals (CIs) for some secondary analyses are wide, indicating possible instability.

9) Compare the performance of CA19-9 and FE-1 individually versus their combined use.

10) Enhance survival plots with confidence bands or annotations (e.g., median survival times, sample sizes at risk).

11) Due to the insidious onset, rapid development, and poor prognosis, PC is a serious threat to human health. Therefore, there is a dire need to diagnose early-stage PC using an effective method, which facilitates in reducing the mortality and improving patient survival rates. Following reference could be added: doi: 10.4251/wjgo.v16.i4.1256.

Author Response

1)The cutoff values for CA19-9 (385 IU/mL) and FE-1 (100 µg/g) are pivotal to the analysis but appear derived primarily from internal data or prior literature. Clarify how confounders were adjusted in the multivariate analysis. Did the team perform sensitivity analyses to ensure the robustness of cutoff values (e.g., CA19-9 at 385)?

: Thank you for your thoughtful questions and valuable feedback. I agree with your opinion. The same trend was observed in cox-proportional regression analysis performed with an FE-1 cutoff of 200. This could result in ensuring the robustness.

[<100 mcg/g (Severe Pancreatic Insufficiency), 100-200 mcg/g (Moderate Pancreatic Insufficiency), >200 mcg/g (Normal) ]

5-year OS

1-year RFS

HR (95% CI)

p-value

HR

p-value

Age

< 58

1

ref.

1

ref.

≥ 58 and < 66

0 (0, 0)

0.000

0 (0, 0)

0.000

≥ 66 and < 73

0 (0, 0)

0.000

0 (0, 0)

0.000

≥ 73

0.77 (0.52, 1.14)

0.187

0.82 (0.56, 1.21)

0.323

Sex

Male

1

ref.

1

ref.

Female

0.97 (0.74, 1.25)

0.789

1.04 (0.79, 1.37)

0.774

T classification

T1

1

ref.

1

ref.

T2

1.47 (0.9, 2.41)

0.120

2 (1.08, 3.71)

0.027

T3/4

1.4 (0.82, 2.4)

0.221

2.11 (1.1, 4.05)

0.026

N classification

N0

1

ref.

1

ref.

N1

1.39 (1.01, 1.93)

0.044

1.4 (1, 1.96)

0.048

N2

2.37 (1.63, 3.46)

<.0001

1.74 (1.15, 2.62)

0.008

Perineal Invasion

No

1

ref.

1

ref.

Yes

0.67 (0.41, 1.1)

0.116

1.1 (0.67, 1.81)

0.710

Lymphovascular invasion

No

1

ref.

1

ref.

Yes

0.83 (0.63, 1.08)

0.162

0.84 (0.64, 1.12)

0.243

Cell differentiate

Well

1

ref.

1

ref.

Moderate

1.97 (1.11, 3.5)

0.021

3.03 (1.4, 6.52)

0.005

Poorly or undifferentiated

2.94 (1.56, 5.55)

0.001

6.18 (2.77, 13.82)

<.0001

Resection Margine, N (%)

R0

1

ref.

1

ref.

R1

0.95 (0.66, 1.39)

0.805

1.17 (0.79, 1.72)

0.442

Preoperative CA19-9 and FE-1

CA19-9<385 and FE-1≥200

1

ref.

1

ref.

CA19-9<385 and FE-1<200

1.18 (0.84, 1.66)

0.341

1.07 (0.74, 1.55)

0.713

CA19-9≥385 and FE-1≥200

1.32 (0.86, 2.03)

0.204

1.86 (1.2, 2.86)

0.005

CA19-9≥385 and FE-1<200

1.84 (1.28, 2.64)

0.001

2.04 (1.4, 3)

0.000

2) The manuscript does not mention whether the proportional hazards (PH) assumption was tested for the Cox models.

: Thank you for your thoughtful questions and valuable feedback.  We described this analysis in the methods section.

In methods section,

‘Univariate and multivariate analyses were performed using the Cox proportional hazard regression model to determine the effects of different variables on survival. The multivariate model was adjusted for age (<58, ≥58 and <66, ≥66 and <73, or ≥73 years), sex (male or female), T classification (T1, T2, or T3/T4), N classification (N0, N1, or N2), perineural invasion (negative or positive), lymphovascular invasion (negative or positive), histologic differentiation (well, moderate, or poorly/undifferentiated), resection margin status (R0 or R1), and combined CA19-9 and FE-1 levels as covariates.’

3) Multivariate analysis combines CA19-9, FE-1, and other clinicopathological variables, but multicollinearity between these predictors is not addressed.

: Thank you for your thoughtful questions and valuable feedback .

In response to the reviewer’s comment, we checked Variance inflation (VIF) to investigate whether there was multicollinearity among the independent variables hypothesized to affect outcomes. As a result of additional analysis, the VIF of all variables was observed to be less than 1.5. As we confirmed that there were no variables with a VIF of more than 10, we judged that our research model did not have high multicollinearity and did not modify it.

The VIF analysis results are as follows.

Variables

VIF

Age

1.02

Sex (male or female)

1.03

T stage (T1, T2, or T3&4)

1.17

N stage (N0, N1, or N2)

1.27

Perineural invasion (no or yes)

1.10

Lymphovascular invasion (no or yes)

1.17

Histologic differentiate (Well, moderate, or poorly/undifferentiated)

1.03

Resection Margine (R0 or R1)

1.06

Preoperative CA19-9 and FE-1 level (Group 1, Group 2, Group 3, or Group 4)

1.07

Group 1: CA19-9 below and FE-1 above the cutoff values (CA19-9 (385 IU/mL) and FE-1 (100 µg/g)); Group 2: both CA19-9 and FE-1 below the cutoff values; Group 3: both CA19-9 and FE-1 above the cutoff values; and Group 4: CA19-9 above and FE-1 below the cutoff values

4) Differences in baseline characteristics between the four groups based on CA19-9 and FE-1 cutoffs are mentioned (e.g., age, tumor stage). However, it is unclear if these differences were adjusted in survival analyses.

: Thank you for your thoughtful questions and valuable feedback. In this analysis, matching analysis was not performed because matching analysis cannot be performed when there are more than two groups. Therefore, adjusted multivariate analysis was used to confirm the results.

5) Subgroup analyses, particularly for small-sized tumors (≤2 cm), may suffer from limited sample size.

: Thank you for your thoughtful questions and valuable feedback. Generally, the number of patients with pancreatic tumors smaller than 2 cm is limited, which represents a limitation of this study. However, since the study is ongoing, further research may be possible as the patient population increases. This has been mentioned in the Discussion section.

‘Third, the number of patients with pancreatic tumors smaller than 2 cm is generally limited. However, as the study is ongoing, further research may be possible as the patient population increases.’

6) The area under the curve (AUC) for CA19-9 is reported as 0.61, which is modest and suggests limited discriminatory ability.

: Thank you for your thoughtful questions and valuable feedback . Your point is correct. Because of these problems, this study suggests that a new risk factor combining CA19-9 and FE-1 is meaningful. This can complement the shortcomings of a single marker.

7) Numerous p-values are presented without adjustment for multiple comparisons.

: Thank you for your thoughtful questions and valuable feedback. All values ​​used in the analysis were necessary to help readers understand. If there is anything that needs to be deleted for the validity of the paper, I will revise it. I would appreciate it if the editor could give me some feedback.

8) Hazard ratios (HRs) are provided, but confidence intervals (CIs) for some secondary analyses are wide, indicating possible instability.

: Thank you for your thoughtful questions and valuable feedback. The results of this analysis were obtained due to the small sample size in this study. Additional accurate analysis is needed in the future with a larger number of patients.

:

9) Compare the performance of CA19-9 and FE-1 individually versus their combined use.

: Thank you for your thoughtful questions and valuable feedback . I will provide the rationale for why combining two markers represents a better predictive strategy. The following content has been added to the introduction section.

‘CA19-9 can be difficult to measure due to its dependence on Lewis antigens and is also influenced by inflammation at the time of measurement. It has the disadvantage of requiring invasive procedures, making frequent measurements challenging. Given these limitations as a single marker, I believe that combining it with other risk factors to create a new marker holds significant potential.’

10) Enhance survival plots with confidence bands or annotations (e.g., median survival times, sample sizes at risk).

: Thank you for your thoughtful questions and valuable feedback. As you advised, I re-created survival plots by adding more abundant information (confidence intervals and median survival lines). We are resubmitting Figures 1 and 2 accordingly.

Figure 1. Kaplan-Meier survival curve by groups on based preoperative CA19-9 and FE-1 level. (a) 3-year overall survival (P < 0.001) and (b) 1-year recurrence free survival (P < 0.001)

Group 1: CA19-9 below and FE-1 above the cutoff values; Group 2: both CA19-9 and FE-1 below the cutoff values; Group 3: both CA19-9 and FE-1 above the cutoff values; and Group 4: CA19-9 above and FE-1 below the cutoff values

Figure 2. Kaplan-Meier survival curve for by groups on based preoperative CA19-9 and FE-1 level on patients with tumor size ≤ 2cm

(a) 3-year overall survival (P < 0.001) and (b) 1-year recurrence free survival (P < 0.001)

Group 1: CA19-9 below and FE-1 above the cutoff values; Group 2: both CA19-9 and FE-1 below the cutoff values; Group 3: both CA19-9 and FE-1 above the cutoff values; and Group 4: CA19-9 above and FE-1 below the cutoff values

11) Due to the insidious onset, rapid development, and poor prognosis, PC is a serious threat to human health. Therefore, there is a dire need to diagnose early-stage PC using an effective method, which facilitates in reducing the mortality and improving patient survival rates. Following reference could be added: doi: 10.4251/wjgo.v16.i4.1256.

: Thank you for your thoughtful questions and valuable feedback. I added a reference. Thank you for introducing such a good journal. This reference has made our study even better. This has been mentioned in the Introduction section.

‘Early diagnosis using effective methods is crucial to reduce mortality and improve survival rates6.’

Reviewer 2 Report

Comments and Suggestions for Authors

In this article, the authors performed the study based on a cohort of pancreatic cancer patients. They detected the role of fecal elastase as a predictive marker in combination with CA19-9. There are few questions need to be answered as listed below:

1.      Does the study perform the analysis based on only FE-1 or CA19-9 using the patients’ information? It is better to provide the results and demonstrate if FE-1 combined with CA19-9 is a better predictive strategy compared with single predictor.

2.      The authors determined the CA19-9 cutoff value based on the study cohort and performed the analysis. An additional patient group is necessary to validate the cutoff value of CA19-9 and FE-1 to support the authors’ conclusion.  

3.      Please provide the reference when cite the study on FE-1 cutoff value (line 99, line 231).

4.      The figure legends of figures 1 and 2 are not very clear. Please describe each panel in more detail and include the p value.

5.      Please check whether it is DFS or RFS in table 3. The main text is RFS but table 3 shows DFS.

Author Response

1.Does the study perform the analysis based on only FE-1 or CA19-9 using the patients’ information? It is better to provide the results and demonstrate if FE-1 combined with CA19-9 is a better predictive strategy compared with single predictor.

: Thank you for your thoughtful questions and valuable feedback. I will provide the rationale for why combining two markers represents a better predictive strategy. The following content has been added to the introduction section.

‘CA19-9 can be difficult to measure due to its dependence on Lewis antigens and is also influenced by inflammation at the time of measurement. It has the disadvantage of requiring invasive procedures, making frequent measurements challenging. Given these limitations as a single marker, I believe that combining it with other risk factors to create a new marker holds significant potential.’

2.The authors determined the CA19-9 cutoff value based on the study cohort and performed the analysis. An additional patient group is necessary to validate the cutoff value of CA19-9 and FE-1 to support the authors’ conclusion.  

: Thank you for your thoughtful questions and valuable feedback . This is our limitation. Therefore, further research is needed through external validation in the future. It is described in the Discussion section.

‘Further research is required for external validation in the future’

3.Please provide the reference when cite the study on FE-1 cutoff value (line 99, line 231).

: Thank you for your thoughtful questions and valuable feedback. I added references.

4.The figure legends of figures 1 and 2 are not very clear. Please describe each panel in more detail and include the p value.

: Thank you for your thoughtful questions and valuable feedback. I have modified it in detail as you mentioned.

Figure 1. Kaplan-Meier survival curve by groups on based preoperative CA19-9 and FE-1 level

(a) 3-year overall survival (P < 0.001) and (b) 1-year recurrence free survival (P < 0.001)

Group 1: CA19-9 below and FE-1 above the cutoff values; Group 2: both CA19-9 and FE-1 below the cutoff values; Group 3: both CA19-9 and FE-1 above the cutoff values; and Group 4: CA19-9 above and FE-1 below the cutoff values

Figure 2. Kaplan-Meier survival curve for by groups on based preoperative CA19-9 and FE-1 level on patients with tumor size ≤ 2cm

(a) 3-year overall survival (P < 0.001) and (b) 1-year recurrence free survival (P < 0.001)

Group 1: CA19-9 below and FE-1 above the cutoff values; Group 2: both CA19-9 and FE-1 below the cutoff values; Group 3: both CA19-9 and FE-1 above the cutoff values; and Group 4: CA19-9 above and FE-1 below the cutoff values

5.Please check whether it is DFS or RFS in table 3. The main text is RFS but table 3 shows DFS.

: Thank you for your thoughtful questions and valuable feedback. I have corrected the points you mentioned. Table 3 has been revised.

Round 2

Reviewer 2 Report

Comments and Suggestions for Authors

Questions are answered. No further comments.